# EFFICIENT COMMUNICATIONS IN TRAINING LARGE SCALE NEURAL NETWORKS

**Linnan Wang**
School of Computer Science
Georgia Institute of Technology
linnan.wang@gatech.edu

**Wei Wu & George Bosilca**
Innovative Computing Laboratory
The University of Tennessee, Knoxville
{wwu12, bosilca}@icl.utk.edu

**Richard Vuduc**
School of Computational Science & Engineering
Georgia Institute of Technology
richie@cc.gatech.edu

**Zenglin Xu**
Big Data Research Center
Univ. of Electr. Sci. & Tech. of China
zlxu@uestc.edu.cn

## ABSTRACT

We consider the problem of how to reduce the cost of communication that is required for the parallel training of a neural network. The state-of-the-art method, Bulk Synchronous Parallel Stochastic Gradient Descent (BSP-SGD), requires many collective communication operations, like broadcasts of parameters or reductions for partial gradient aggregations, which for large messages quickly dominates overall execution time and limits parallel scalability. To address this problem, we develop a new technique for collective operations, referred to as Linear Pipelining (LP). It is tuned to the message sizes that arise in BSP-SGD, and works effectively on multi-GPU systems. Theoretically, the cost of LP is invariant to $P$, where $P$ is the number of GPUs, while the cost of the more conventional Minimum Spanning Tree (MST) scales like $O(\log P)$. LP also demonstrates up to 2x higher bandwidth than Bidirectional Exchange (BE) techniques that are widely adopted by current MPI implementations. We apply these collectives to BSP-SGD, showing that the proposed implementations reduce communication bottlenecks in practice while preserving the attractive convergence properties of BSP-SGD.

## 1 INTRODUCTION

Scaling up neural networks with respect to parameter sizes, training sets, or both has drastically improved the state-of-the-art performance in several domains ranging from scene understanding, speech recognition, even to playing Go against professional players. Although training a large network saturated with nonlinearities is extremely time-consuming, the benefits brought forth by large-scale models has sparked a surge of interest in parallelizing training on multi-GPUs. The parallelization of SGD demands synchronizations to exchange gradients and parameters per iteration, and this introduces significant communication overhead. Previous studies have focused on trading the SGD convergence rate for fast gradient updates, such as stale or asynchronous SGD, 1-bit compressed gradient, etc. However, these methods are rarely adopted by Deep Learning frameworks as they depend on the balance between the enhanced iteration throughput and the decelerated convergence rate. Since BSP retains the convergence properties of SGD, its optimization should be of interest.

The gradient aggregations and parameter exchanges in BSP SGD are typical operations of communication collectives (Chan et al., 2007). Messages in the large-scale neural networks training are dense, long, and fixed-length, while the performance of collective algorithms is drastically sensitive to these attributes. Besides, the processing speed is several orders of magnitude faster than the network unidirectional transmission rate. These prioritize the utilization of network bandwidth in the collective design. However, we have seen sub-optimal collective algorithms, e.g. MST and BE, widely adopted by the deep learning community (Agarwal et al., 2014) (Jia et al., 2014) (Duchi et al., 2011). MST is only suitable for the latency dominant case such as frequent short message exchanges, while the bandwidth term of BE can be further improved (Thakur et al., 2005).

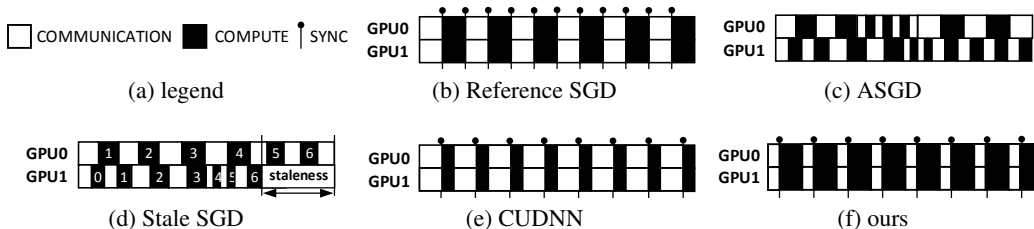

Figure 1: Illustrations of various methods to accelerate the training. Black blocks stands for computations, and white blocks stands for communications. CUDNN reduces the computation cost, while we reduce the communication cost.

In this paper, we introduce new Linear Pipeline based collectives for multiGPU training. The collectives demonstrate $\mathcal{O}(log(P))$ speedups over MST collectives and up to 2x speedups over BE based ones; the bounds only hold in training large neural networks. In particular, the theoretical analysis and the implementation yield an interesting insight that ***the cost of our design is invariant to GPU numbers***, i.e., the cost of collective operations on 2 GPUs is similar to 20 GPUs. The design explores message granularity to maximize simultaneous bidirectional data exchanges. In specific, it divides a message into fine-grained blocks as the basic communication element. A GPU sends a block (via DMA 1) while receiving (via DMA 2) a new block from a neighbor. The copies are asynchronously launched on two GPU streams, and numerical operations further overlap data copies. As a result, our method yields a highly efficient pipeline over which messages for neural network training may be exchanged.

The proposed collective design achieves 2.3x to 360.55x speedups over Open MPI alternatives on 6 GPUs. In training GoogLeNet, we set up the same BSP SGD implementation with different underlying collectives. Our design demonstrates up to 1.7x convergence speedup over MST based Caffe.

## 2 RELATED WORK

The communication overhead has been widely identified as the major bottleneck in the data-parallel SGD (Shamir (2014), Li et al. (2014)). The data parallelism linearly adds the processing power by concurrent gradient computations with multiple GPUs. But it also requires synchronizations to collect partial gradients or to broadcast parameters. In practice, the communication rate is several orders of magnitude slower than the computation (Coates et al., 2013). Various approaches have been proposed to reduce the overhead.

The first group of approaches relaxes synchronous models of SGD to increase the iteration throughput (Dean et al. (2012), Zinkevich et al. (2010)). In this case, the relaxed SGD enables computations on a GPU to partially overlap with communications on others as demonstrated in Fig.1c and Fig.1d. Recht et al. (2011) proposed a lock free Asynchronous SGD (ASGD) that entirely gets rid of the synchronization requirement by allowing free concurrent parameter updates. But the relaxation only works well on sparse learning problems. In response, Ho et al. (2013) introduced the concept of staleness by bounding the fastest and the slowest machine within a few iterations of each other to ensure correctness. These relaxations claim to be effective as the enhanced iteration throughput offsets the disadvantages of degraded convergence rate. However, recent advances in deep learning frameworks (Cui et al. (2016)) have reestablished the advantages of BSP over relaxed ones in training neural networks. This reiterates the importance of studying BSP SGD.

The second group of approaches tries to reduce the overall communication volume. Seide et al. (2014) quantized gradients from 32 bits to 1 bit to reduce the message length, but the lost gradient information decelerates the convergence rate. Another approach is to accelerate the convergence with a large batch. Dekel et al. (2012) shows the convergence rate of mini-batch SGD is $\mathcal{O}(1/\sqrt{Tb} + 1/T)$ with $b$ being the batch size. This result indicates a large batch needs fewer iterations to find a solution, and thereby fewer overall synchronizations. However, unwieldy increasing the batch size is also unfavorable under limited computing resources demonstrated by Wang et al. (2016b). Please note these methods still need synchronizations, and our work will further improve their performance.

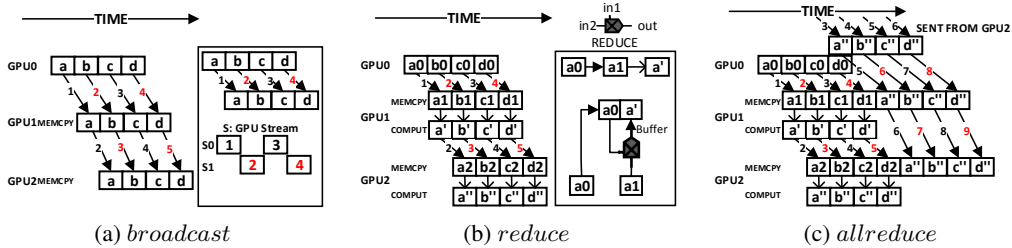

(a) *broadcast*    (b) *reduce*    (c) *allreduce*

Figure 2: The data flow of *broadcast*, *reduce* and *allreduce* on 3 GPUs.

The third group of approaches conducts system optimizations to minimize the communication cost (Wang et al., 2016a). Agarwal & Duchi (2011) and Agarwal et al. (2014) presented partial gradients aggregations guided with a MST that takes $log(P)$ steps to fully synchronize the model. Deep learning frameworks such as Caffe (Jia et al., 2014) also adopt this approach. Unfortunately, MST is only suitable for latency dominant scenarios (i.e. high frequent short messages). Although collective algorithms have been thoroughly discussed in the HPC community (Almási et al. (2005), Gabriel et al. (2004), Shipman et al. (2006)), few have studied their performances for the deep learning. The performance of collectives varies significantly with different message lengths and network topologies, while messages in deep network training are dense, long and fixed-length. Therefore, it is imperative to address such peculiarities in the collectives. Worringen (2003) proposed a pipeline collective model in shared memory environment for CPU data, but communications of different MPI processes sharing the same CPU memory bus within the same CPU socket. This causes bandwidth competition among different processes, thereby poor performance for the collective communication in shared memory environment for CPU data. In contrast, PCI-E is bi-directional. The latest GPUs also feature two independent DMA engines for simultaneous independent in/out communications. The hardware updates pave the way for LP based GPU communications.

## 3 LINEAR PIPELINE BASED COLLECTIVE DESIGN DEDICATED FOR NEURAL NETWORK TRAINING ON MULTI-GPUS

This section presents a new LP based MultiGPU collective design ensued by the concrete proof of its performance in training neural networks. The general idea of LP is as follows: a) we dissect a long message into fine-grained blocks. b) a GPU receives a block from the prior GPU via DMA1 while sending a block to the next one via DMA2. Please note each block exchange utilizes an independent physical link, and the entire network is fully utilized once the pipeline is filled.

***Broadcast*** tackles the synchronizations of parameters among multiple GPUs. It copies the source vector to every GPU. Fig.2a illustrates the data flow of the *broadcast* collective on 3 GPUs. GPU0 is the source, and the rest are destinations. *Broadcast* starts with filling the pipe by copying block $a$ on GPU0 to GPU1 at step 1. Let's focus on GPU1. At each step, GPU1 receives a block from GPU0 via DMA1, while GPU1 is also sending a block to GPU2 via DMA2. The data exchange in either way utilizes an independent link and DMA engine to achieve the maximal unidirectional rate. Hence, the bandwidth is fully exploited.

***Reduce*** aggregates the partial gradients to reconstruct the global one. It combines the elements provided in the vector of each GPU, and returns the combined value in the receive vector to a specific GPU. It supports basic arithmetic operations such as summations and multiplications. Fig.2b illustrates the data flow of the *reduce* collective. GPU2 is the root that aggregates the vectors across all GPUs. *Reduce* starts with filling the pipe by writing block $a0$ to a buffer on GPU1. Then, GPU1 reduces the received block $a0$ with $a1$ to yield $a'$ (within the rectangle of Fig.2b). Please note the computation is much faster than the communication, we assume no latency on it. In practice, computations are further overlapped with communications. In the next step, GPU1 retrieves $b0$ from GPU0 to reduce to $b'$ via DMA 1, while GPU1 is also sending $a'$ to GPU2 to reduce to $a''$ via DMA 2. $b''$, $c''$, $d''$ are reduced at steps 3, 4, 5 in a similar fashion.

***AllReduce*** enables us to collect partial gradients and broadcast the latest parameters with only one synchronization point per SGD iteration. It combines vectors from all GPUs and distributes the

Table 1: The estimated costs of 3 collective communications.

| | Bidirectional Exchange (BE) | Minimal Spanning Tree (MST) | Linear Pipeline (LP) |
|---|---|---|---|
| $broadcast$ | $(\log p + p - 1)\alpha + 2(\frac{p-1}{p}n)\beta$ | $\log p(\alpha + n\beta)$ | $(p-1+\frac{n}{b})\alpha + (b(p-1)+n)\beta$ |
| $reduce$ | $(2\log p)\alpha + 2(\frac{p-1}{p}n)\beta + (\frac{p-1}{p}n)\gamma$ | $\log p(\alpha + n\beta + n\gamma)$ | $(p-1+\frac{n}{b})\alpha + (bp-b+n)(\beta+\gamma)$ |
| $allreduce$ | $(2\log p)\alpha + 2(\frac{p-1}{p}n)\beta + (\frac{p-1}{p}n)\gamma$ | $\log p(2\alpha + 2n\beta + n\gamma)$ | $2(p-1+\frac{n}{b})\alpha + (bp-b+n)(2\beta+\gamma)$ |

result back to them. Mathematically, it is equivalent to a $reduce$ followed by a $broadcast$. However, $allreduce$ is more efficient than two separate calls as it only needs to fill the pipeline once. For example, it takes 9 timesteps to $allreduce$ 4 message blocks, while $broadcast + reduce$ will cost 10. Fig.2c illustrates the data flow of the $allreduce$ collective. It starts with reducing $a''$, after which $a''$ is broadcast to GPU1 and GPU2 at step 5, 6 respectively. Please note $d0$ utilizes the outbound DMA at step 4, therefore $a''$ has to wait until step 5. $b''$, $c''$, $d''$ are processed in a similar fashion.

Our collective is also specifically designed to accommodate GPU features such as asynchronous kernel launches and multi-stream processing. In the rectangle of Fig.2a, it demonstrates the data transfers are asynchronously launched on two separate streams. The copies happening in the red steps are scheduled on one stream while copies in the black steps are scheduled on another stream. This overlaps the overhead of GPU kernel launches, further improving the pipeline. We illustrate the data flow of the collectives on 3 GPUs. If there are $k$ GPUs, GPU $n$, $0 < n < k - 1$, duplicates the same communication pattern on GPU 1.

## 3.1 Architecture Analysis

LP is the optimal collective algorithm to fully exploit the network bandwidth of a MultiGPU system. Even though PCI-E supports full-duplex communication between any two endpoints, each PCI-E endpoint device only has one input and output port. This results in bandwidth competition if a GPU is receiving from multiple GPUs. Similarly, each PCI-E switch only contains one input and output port used for inter-switch communication, and inter-switch communications of the same direction also compete for the PCI-E bus. It is known that any delay in data movement between two GPUs interrupts the pipelining in the collectives. In such architecture, the communication from parents to children in MST based collective algorithms will compete for the same PCI-E bus, therefore breaking pipelining. The data exchange of BE also suffers from the inter-switch communication congestion in one direction. In contrast, LP connects all GPUs into a chain, and data always flow in one direction. Hence, data movements between two GPUs exclusively occupy the entire PCI-E bus, ensuring uninterrupted pipelining.

## 3.2 Theoretical Analysis

We adopt a cost model widely used by the MPI community to analyze collective operations (Thakur et al. (2005), Thakur & Gropp (2003)). The model assumes the time taken to send a message between two nodes follows:

$$T = \alpha + \beta n + \gamma n \qquad (1)$$

where $\alpha$ is the latency or startup time of sending a message, $\beta$ and $\gamma$ is the transmission rate and reduce rate measured by time per byte, and n is the message size in bytes. We also denote $p$ as the node count, and $b$ as the block size (in bytes) in the pipeline.

**Proposition 1** *If the network latency $\alpha \to 0$, Linear Pipeline collectives provide an $\mathcal{O}(\log p)$ speedup over Minimal Spanning Tree collectives and up to a 2 times speedup over Bidirectional Exchange collectives as the message size $n \to \infty$.*

***Proof.*** First, we derive the costs of the three Linear Pipeline collectives. According to Fig.2, the length of pipeline is $p - 1 + \frac{n}{b}$ blocks assuming each block to be $b$ bytes. A block exchange takes $\alpha + \beta b + \gamma b$ (with reduce) or $\alpha + \beta b$ (without reduce). Consequently, $broadcast$ essentially costs $(\alpha+\beta b)(p-1+\frac{n}{b}) = (p-1+\frac{n}{b})\alpha + (b(p-1)+n)\beta$, and $reduce$ costs $(\alpha+\beta b+\gamma b)(p-1+\frac{n}{b}) = (p - 1 + \frac{n}{b})\alpha + (b(p - 1) + n)(\beta + \gamma)$. $allreduce$ is approximately equivalent with a $reduce$

followed by a $broadcast$. Therefore, the $allreduce$'s cost is $broadcast$'s cost plus $reduce$'s cost, i.e. $2(p - 1 + \frac{n}{b})\alpha + (bp - b + n)(2\beta + \gamma)$.

Secondly, we derive the costs of the three Minimal Spanning Tree collectives. MPI adopts MST to $broadcast$ or $reduce$ short messages (Thakur et al. (2005)), the length of which is less than 12 KB. The core concept of MST is to organize $p$ GPUs into a balanced tree of height $\lceil logp \rceil$. Then, it takes $\lceil \log p \rceil$ steps to traverse all GPUs in the tree. Each step carries the message of length $n$, resulting in the cost of $broadcast$ to be the tree height times the cost per step, i.e. $\log p(\alpha + n\beta)$ (we omit the ceiling for simplicity). Similarly, MST $reduce$ is $\log p(\alpha + n\beta + n\gamma)$, and MST $allreduce$ is also a combination of $broadcast$ and $reduce$. Please note the latency term, $\log p\alpha$, is the smallest among algorithms in Table.1, and the bandwidth term, $\log pn\beta$, is the slowest as $\log pn\beta \gg n\beta$. Therefore, MST is widely used for high frequent exchanges of short message.

Finally, we present the costs of the three Bidirectional Exchange collectives. MPI $broadcast$ handles long messages with a MST $scatter$ followed by a BE $allgather$. Please refer to Chan et al. (2007) for the analysis of BE collectives. Basically, $scatter$ costs $\sum_{k=1}^{\lceil logp \rceil}(\alpha + 2^{-k}n\beta) = \log p\alpha + \frac{p-1}{p}n\beta$, while $allgather$ costs $(p - 1)\alpha + \frac{p-1}{p}n\beta$. The cost of $broadcast$ is the sum of these two. The MPI long message $reduce$ consists of a $reducescatter$ plus a $gather$, while $allreduce$ consists of a $reducescatter$ and a $allgather$. The cost for $reducescatter$ is $\log p\alpha + \frac{p-1}{p}n\beta + \frac{p-1}{p}n\gamma$, and both the costs of $gather$ and $allgather$ are $\log p\alpha + \frac{p-1}{p}n\beta$ (also in Chan et al. (2007)). Table 1 summarizes the costs of $broadcast$, $reduce$ and $allreduce$ for the three different underlying algorithms.

The proposition holds under the assumptions of $\alpha \to 0$ and $n \to \infty$, and these assumptions are legitimate for the training of large scale neural networks on multiGPUs. Nowadays, the PCI Express x16 effectively reduces the latency $\alpha$ down to $10^{-7}s$. The current two sockets shared memory machine supports up to 8 GPUs indicating limited $p$ in practice. Let's take an appropriate block size $b$ to ensure $p \ll \frac{n}{b}$ and $\alpha\frac{n}{b} \sim 0$. This enables us to safely ignore the latency term, e.g. $\log p\alpha$ in MST $broadcast$. On the other hand, current deep convolutional neural network uses a tremendous number of parameters. For example, AlexNet uses 50 MB parameters. The transmission rate[1] $\beta \sim 10^9 Byte/Seconds$. Compared to the trivial latency term, the bandwidth term dominates the entire cost $T$. This result leads us to simplify the costs of BE, MST, and LP based $broadcast$ (Table. 2) to be $2\frac{p-1}{p}n\beta$, $n\beta \log p$ and $(b(p - 1) + n)\beta$, obtaining the following equations:

$$\frac{T_{broadcast\_BE}}{T_{broadcast\_LP}} \approx \frac{2(1 - \frac{1}{p})}{1 + \frac{b}{n}(p - 1)} < 2 \tag{2}$$

$$\frac{T_{broadcast\_MST}}{T_{broadcast\_LP}} \approx \frac{\log p}{\frac{b(p-1)}{n} + 1} < \log p \tag{3}$$

Compared with $broadcast$, $reduce$ has the additional $\gamma$ term. Please note the processing speed of GPUs exceeds TFLOPs implying the term $\gamma * n \to 0$. Therefore, it is also legitimate to ignore the $\gamma$ term, and it yields the same result $T_{reduce\_BE}/T_{reduce\_LP} < 2$ and $T_{reduce\_MST}/T_{reduce\_LP} < \log p$. This completes our proof of the proposition 1.

Another interesting point is the cost of *Linear Pipeline is invariant to GPU count $p$ regardless of message length* $n$. This implies broadcasting a vector to 8 GPUs should cost the same as broadcasting to 2 GPUs. In practice, we set the block size $b$ around 64 KB, and $p$ is within $10^1$. This suggests the bandwidth term, e.g. the cost of LP $broadcast$ $(bp - p + n)\beta \sim n\beta$. Hence, the cost of LP collectives are less likely to be affected by GPU counts $p$.

## 3.3 DEEP LEARNING WITH EFFICIENT BSP SGD

We formulate the neural network training as the following optimization problem. Let $\psi$ be a loss function with weight vector $\mathbf{w}$ as function parameters that takes randomly sampled images $\mathbf{d_t}$ as the

---

[1] https://en.wikipedia.org/wiki/InfiniBand

---

**Algorithm 1:** BSP SGD with communications/computations overlapping.

1  **while** *not converge* **do**
2 $broadcast(\mathbf{w}_t^0)$
3 **for** $i \in [0, 1, ..., max\_layers]$ **do**
4 $nonblocking\_broadcast(\mathbf{w}_t^{i+1})$
5 Forward($i$)
6 $sync\_broadcast()$
7 Backward($max\_layers$)
8 **for** $i \in [max\_layers - 1, ..., 1, 0]$ **do**
9 $nonblocking\_reduce(\nabla\psi_{\mathbf{sub}}^{\mathbf{i+1}})$
10 Backward($i$)
11 $sync\_reduce()$
12 $\mathbf{w_{t+1}}$ = GradientUpdate()

---

**Algorithm 2:** BSP SGD uses $broadcast + reduce$.

1  **while** *not converge* **do**
2 $\nabla\psi_{sub}$ = ForwardBackward($\mathbf{d_t}$)
3 $\nabla\psi = reduce(\nabla\psi_{sub})$
4 **if** *root* **then**
5 $\mathbf{w_{t+1}}$ = GradientUpdate()
6 $broadcast(\mathbf{w_{t+1}})$
7 $barrier$ /* sync new $\mathbf{w}$ */

**Algorithm 3:** BSP SGD uses allreduce.

1  **while** *not converge* **do**
2 $\nabla\psi_{sub}$ = ForwardBackward($\mathbf{d_t}$)
3 $\nabla\psi = allreduce(\nabla\psi_{\mathbf{sub}})$
4 $barrier$ /* collect $\nabla\psi_{sub}$ */
5 $\mathbf{w_{t+1}}$ = GradientUpdate()
6 **if** $iter\%5 = 0$ **then**
7 $broadcast(\mathbf{w_{t+1}})$

---

input. The objective of training is to find an approximate solution to the following problem:

$$\min_{\mathbf{w}} \quad E\{\psi_{\mathbf{w}}(\mathbf{d_t})\} = \int_{\Omega} \psi_{\mathbf{w}}(\mathbf{d_t})dP \tag{4}$$

A typical neural network training iteration consists of a forward and backward pass. The forward pass yields a loss that measures the discrepancy between the current predictions and the target; The backward pass calculates the gradient, the negative of which points to the steepest descent direction. The gradient descent updates the parameters, $\mathbf{w}$, as follows:

$$\mathbf{w}^t = \mathbf{w}^{t-1} - \eta_t \nabla\psi_{\mathbf{w}}(\mathbf{d_t}) \tag{5}$$

Guided with Data Parallelism, BSP SGD evenly divides $\mathbf{d_t}$ into $p$ slices $\mathbf{d_t^1}, \mathbf{d_t^2}, ..., \mathbf{d_t^P}$ so that every GPU computes a partial gradient from $\mathbf{d_t^i}$ in parallel. The global gradient is equivalent to the average of partial gradients. After finishing the gradient update, $\mathbf{w}^t$ is synchronized to all GPUs. We integrate the proposed collectives into this process to harness parallel processing capabilities of multiGPU system. In this paper, we discuss two approaches to BSP SGD implementations.

•**fork and join**: This approach forks the gradient computations, and joins partial gradients with communications. In this case, communications do not overlap with computations. Alg.2 and Alg.3 demonstrate two collective based implementations using 2 and 1 synchronization points, respectively.

In Alg.2, synchronizations rely on $broadcast$ and $reduce$. Each GPU calculates a partial gradient referred to as $\nabla\psi_{sub}$. The master GPU reconstructs $\nabla\psi$ by reducing all $\nabla\psi_{sub}$. Then, the GPUs synchronize the latest weight, $\mathbf{w}$, by broadcasting.

In Alg.3, synchronizations only rely on $allreduce$. The differences between this and Alg.2 are that 1) there is only 1 synchronization point; 2) every GPU computes the gradient update. However, the parameters are not consistent after several iterations due to the precision issues of float multiplications in $GradientUpdate$. We synchronize $\mathbf{w}$ every 5 iterations to enforce consistency while still retaining the benefit of efficient pipelining in $allreduce$ (line 7-8 Alg.3).

•**overlapping communications with computations**: Another approach is to overlap communications and computations for each network layer. In the forward pass, GPUs broadcast network parameters of layer t+1 during forward computations at layer t. In the backward pass, GPUs $reduce$

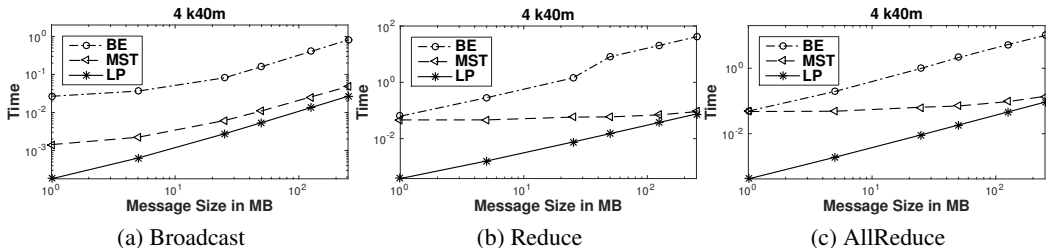

|  |  |  |
|:---:|:---:|:---:|
| (a) Broadcast | (b) Reduce | (c) AllReduce |

Figure 3: The performance of different collective algorithms at different message sizes on 4 K40m.

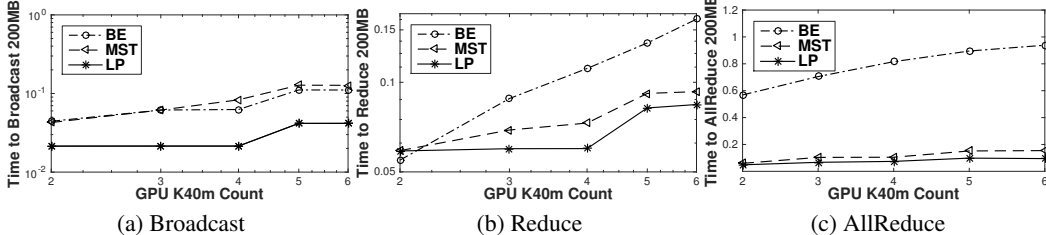

|  |  |  |
|:---:|:---:|:---:|
| (a) Broadcast | (b) Reduce | (c) AllReduce |

Figure 4: The scalability experiment: it measures performance variations with increasing GPUs.

partial gradients of layer t+1 during backward computations at layer t. As a result, layer-wise computations partially overlap with communications further improving the SGD efficiency. Alg.1 outlines the general idea of overlapping communications and computations during network training. We use nonblocking collectives to achieve the overlap.

•**pros and cons of both approaches**: The cost of Alg.2 or Alg.3 is $comm + compt$, while the cost of Alg.1 is $max(comm, compt)$. If the network has over a few hundred MB of parameters, the overlapping will be significantly better than the fork and join approach. However, Alg.2 and Alg.3 are relatively easy to implement, and the performance on networks $< 100$ MB is similar to that of Alg.1.

## 4 EXPERIMENT

### 4.1 COLLECTIVES EVALUATION

The MST and BE implementations used in benchmarks are Caffe [2] and OpenMPI. Caffe optimizes the GPU placement in an MST to fully utilize inter-GPU peer to peer (P2P) access. OpenMPI and our implementation, similar to Caffe, also take advantages of P2P. We set up AlexNet and GoogLeNet training using the three BSP SGD algorithms proposed in section 3.3.

Fig.3 presents the performance of LP, MST, and BE based collectives at different message sizes on 4 K40m. The LP $broadcast$ demonstrates an average of 29.2x and 2.3x speedup over BE and MST based alternatives in Caffe and OpenMPI; the LP $reduce$ demonstrates an average of 360.55x and 8.7x speedup over BE and MST $reduce$, and the LP $allreduce$ demonstrates an average of 109.2x and 7.9x speedup over BE and MST $allreduce$. In theory, LP is approximately 2x faster than both the MST ($p = 4 \rightarrow logp = 2$) and BE approaches. An extraordinary speedup against Open MPI is observable due to inefficient data movement in Open MPI, which moves data to host RAM to perform $reduce$ operations on the CPU before being copied to the target GPU. Instead, we perform $reduce$ on the GPUs, and data blocks directly flow to the target GPU via P2P access. The overlapped $reduce$ computations with communications enables our $reduce$ and $allreduce$ to be 8x faster than that of MST. At each step of MST, GPUs $reduce$ the incoming data only after all the data is available. In contrast, our fine-grained block design enables communications and computations to overlap by reducing a block while receiving a new one in the pipeline. $broadcast$ only involves data copies, and both we and Caffe use P2P to transmit the data. Therefore, the speedup of MST $broadcast$ (2.3x), conforms to the 2.0x theoretical prediction.

The theoretical analysis indicates both the cost of LP and BE collectives are invariant to the GPU count $p$, while the cost of MST increases with $p$ by a factor of $logp$. This is also noticeable in the

---

[2]Caffe implements an MST based broadcast and reduce for the multiGPU training.

Table 2: The iteration profile. comm stands for communications, and compt stands for computations. % represents the percentages of communications in an iteration. The statistics are the average of 30000 AlexNet iterations, and 67000 GoogLeNet iterations. We set the batch size of AlexNet to 1000, and GoogLeNet to 80. AlexNet and GoogLeNet are 256MB and 51MB, respectively.

| | MST Alg.1 | | | BE Alg.1 | | | BE Alg.3 | | | LP Alg.1 | | | LP Alg.2 | | | LP Alg.3 | | |
|---|---|---|---|---|---|---|---|---|---|---|---|---|---|---|---|---|---|---|
| | comm | compt | comm% | comm | compt | % | comm | compt | % | comm | compt | % | comm | compt | % | comm | compt | % |
| AlexNet | 0.77s | 0.92s | 45.5% | 1.05s | 0.94s | 52.7% | 0.22s | 0.93s | 18.9% | 0.084s | 0.93s | 8.3% | 0.057s | 0.94s | 5.7% | 0.011s | 0.93s | 1.2% |
| GoogLeNet | 0.046s | 0.267s | 14.7% | 0.334s | 0.264s | 55.9% | 0.137s | 0.263s | 34.3% | 0.02s | 0.265s | 7% | 0.016s | 0.26s | 5.8% | 0.01s | 0.263s | 3.7% |

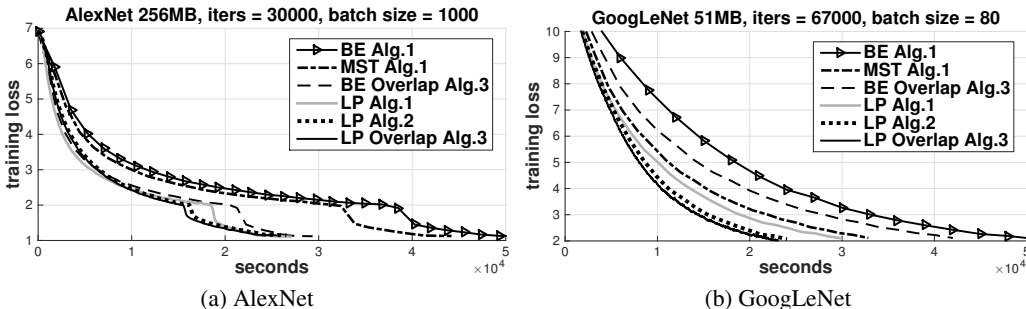

(a) AlexNet    (b) GoogLeNet

Figure 5: The training losses in fixed iterations on 4 K40m. We set GoogLeNet lr = 0.01. AlexNet starts at lr = 0.015, and set to 0.0015 after the average loss $< 2$. The solver is SGD + momentum, and the dataset is ImageNet.

scalability experiment demonstrated in Fig.4. Please note there is a cost jump between 4 and 5 GPUs. Communications have to go through QPI after 4 GPUs incurring the additional cost of copying through the host RAM. The cost of the Linear Pipeline method robustly stays the same if GPU counts =[2,3,4] or [5,6], and QPI explains the inconsistency. The communication steps of MST for 2,3,4,5,6 GPUs are 1,2,2,3,3, respectively. The MST experiments verify the $logp$ cost increase w.r.t GPU counts by evident cost jumps at 3 and 5 GPUs. The data flow of OpenMPI between two GPUs follows GPU RAM→host RAM→GPU RAM. The inefficient data flow inside Open MPI contributes to the near linear cost increase with GPU counts $p$.

## 4.2 IMPACT ON THE NEURAL NETWORK TRAINING

Fig.5 demonstrates LP collectives effectively reduce the total training time without affecting SGD's convergence properties in training large scale neural networks. We use inspurCaffe, Caffe and cuhk's Caffe branch to benchmark the performance of BE-Alg.1, MST-Alg.1 and BE-Overlap-Alg.3. We also implement Alg.1,2,3, integrated with LP collectives, in Caffe to ensure consistency. Please note the model size affects the communication time, while the batch size affects the computation time. We carefully set these parameters to cover as many cases as possible. Please refer to the captions of Table.2 and Fig.5 for experiment details. We assume these algorithms have similar convergence speeds in iterations as losses of AlexNet are approximately 1 after 30000 iterations and losses of GoogLeNet are approximately 2 after 67000 iterations. However, the time taken to reach the target loss varies dramatically. For example, the speedups of LP-Overlap-Alg.3 over BE-Alg.1 in training AlexNet and GoogLeNet are 2.12x and 2.19x, respectively.

Under Alg.1, but using different underlying collective algorithms, LP-Alg.1 presents 1.91x and 1.74x speedup over BE-Alg.1 and MST-Alg.1 in AlexNet, and 1.6x and 1.1x speedup over BE-Alg.1 and MST-Alg.1 in GoogLeNet. The iteration profiles of these 3 algorithms in Table.2 indicate the communication cost of LP-Alg.1 is only 10% of BE-Alg.1, and 11% of MST-Alg.1 in AlexNet; and 6% of BE-Alg.1, and 43% of MST-Alg.1 in GoogLetNet.

The experiments demonstrate that the speed of the three proposed BSP SGD algorithms is Alg.3 > Alg.2 > Alg.1. The result conforms to our expectations as the cost of Alg.3 is $max(comm, compt)$, while the cost of Alg.1 and Alg.2 is $comm + compt$. However, the performance gain is quite limited from Alg.2 to Alg.3 as there is little room left for reducing communications from LP Alg.2 to Alg.3 as demonstrated in Table.2. If the model parameters keep increasing, we expect Alg.3 to be more efficient than Alg.2.

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
