# Peer review of "Efficient Communications in Training Large Scale Neural Networks"

_ICLR 2017 — rejected_

[Official Review · AnonReviewer1 · rating 6 · confidence 4 · 15 Dec 2016]
**Promising approach, but the paper has some problems**

The primary point made by this paper is that given certain architectural characteristics of multi-GPU systems, namely the use of bi-directional PCI-E for communication and the integration of two independent DMA engines on recent GPU devices (providing support for simultaneous independent communications), and given the characteristics of the communications patterns required by synchronous SGD trainers for deep neural networks, namely that the messages are large, dense, and have a fixed length, it makes sense to design communication collectives such as broadcast, reduce, and allreduce specifically for the use case of synchronous SGD training on a multi-GPU system.  The paper describes the implementation of these three collectives (broadcast, reduce, and allreduce) using a linear pipelining (LP) scheme on a (logical) ring topology.  The paper compares the LP collectives to two alternatives:  collectives based on a minimal spanning tree (MST) topology and collectives based on bidirectional exchange (BE).  First, a theoretical comparison is made using a standard cost model used in the high performance computing community.  When assumptions based on multi-GPU system architecture (very low latency for messages) and on the communication characteristics of synchronous SGD training (very large messages) are integrated into the model, the paper finds that the LP collectives should be less costly than BE collectives by a factor of 2 and less costly than MST collectives by a factor of log(p), where p is the number of GPUs being used.  Second, an empirical comparison is performed in which (1) the time required to perform each of the different collectives on a 4-device (k40m) system is measured as a function of message size and (2) the time required to perform each of the different collectives with a 200 MB message length is measured as a function of the number of devices in the system.  These measurements show that the LP-based collectives are consistently the fastest.  Third, DNN training experiments with AlexNet and GoogLeNet are performed on a 4-device system using three different synchronous SGD algorithms with the different implementations of the collectives (a total of 6 different algorithms in all).  Measurements of the communication and computation costs show that the LP collectives reduce communication costs without affecting computation costs (as expected).  Measurements of the convergence of the training loss as a function of time for the two DNN architectures show that use of the LP collectives leads to faster training.

While the theory says that the costs of LP collectives should be invariant to the number of devices in a multi-GPU system, the empirical work shows that in practice this does not hold going from 4 to 5 devices (in the tested configuration) because in a 5-device system messages must traverse the QPI.  Are there other practical considerations that the authors are aware of that affect the scaling of the LP collectives?  If so, these should be mentioned in the paper.

In the sentence "Worringen (2003) proposed a pipeline collective model in shared memory environment for CPU data, but communications of different MPI processes sharing the same CPU memory bus within the same CPU socket." I really can't figure out what the words after "but communications of different MPI processes" are trying to convey.  This sentence is not comprehensible.

"Please note the latency term is log pα, which is the smallest among algorithms in Table.1. Therefore, MST only suits for high frequent short messages."  The claim that MST collectives are only suitable for high-frequency, short messages does not follow from the statement that MST collectives have the smallest latency term.  You also need to consider the way the cost scales with message size (the bandwidth term).  If the MST collectives had a better bandwidth term than the other collectives, then they would also be superior for large messages.

"Let’s take an appropriate block size b to ensure n/b ≪ α."  This looks wrong, since n > b.  Should it be b/n ≪ α?

"However, the parameters are not consistent after several iterations due to the precision issues of float multiplications in Gradient Update."  Are you sure the inconsistency in weight estimates across devices is due to multiplication?  I would expect that it would be due to gradients being accumulated in different orders; that is, because floating point addition is not commutative.

I recommend replacing the term "sub-gradients" in this paper with "partial gradients."  In the optimization literature, the term "sub-gradient" has a very specific meaning that differs from this paper's use of the term (see

[Official Review · AnonReviewer3 · rating 5 · confidence 5 · 16 Dec 2016 (modified: 17 Dec 2016)]
**review for Efficient Communications in Training Large Scale Neural Networks**

This paper analyzes the ring-based AllReduce approach for multi-GPU data parallel training of deep net.
Comments
1) The name linear pipeline is somewhat confusing to the readers, as the technique is usually referred as ring based approach in Allreduce literature. The author should use the standard name to make the connection easier. 
2) The cost analysis of ring-based Allreduce is already provided in the existing literature. This paper applied the analysis to the case of multi-GPU deep net training, and concluded that the scaling is invariant of number of GPUs.
3) The ring-based allreduce approach is already supported by NVidia’s NCCL library, although the authors claim that their implementation comes earlier than the NCCL implementation.
4) The overlap of communication of computation is an already applied technique in systems such as TensorFlow and MXNet. The schedule proposed by the authors exploits the overlap partially, doing backprop of t-1 while doing reduce.  Note that the dependency pattern can be further exploited; with the forward of layer t depend on update of parameter of layer t in last iteration. This can be done by a dependency scheduler.	
5) Since this paper is about analysis of Allreduce, it would be nice to include detailed analysis of tree-shape reduction, ring-based approach and all-to-all approach. The discussion of all-to-all approach is missing in the current paper. 
In summary, this is a paper discussed existing Allreduce techniques for data parallel multi-GPU training of deep net, with cost analysis based on existing results. While I personally find the claimed result not surprising as it follows from existing analysis of Allreduce, the analysis might help some other readers. I view this as a baseline paper. The analysis of Allreduce could also been improved (see comment 5).

[Official Review · AnonReviewer2 · rating 5 · confidence 3 · 20 Dec 2016]
**No Title**

This paper presents a linear pipeline All-reduce approach for parallel neural networks on multiple GPU. The paper provides both theoretical analysis and experiments. Overall, the results presented in the paper are interesting, but the writing can be improved. 

Comments:

- The authors compare their proposed approach with several alternative approaches and demonstrate strong performance of the proposed approaches. But it is unclear if the improvement is from the proposed approach or from the implementation.  

- The paper is not easy to follow and the writing can be improved in many place (aside from typos and missing references). Specifically, the authors should provide more intuitions of the proposed approach in the introduction and in Section 3. 

- The proposition and the analysis in Section 3.2 do not suggest the communication cost of linear pipeline is approximately 2x and log p faster than BE and MST, respectively, as claimed in many places in the paper. Instead, it suggests LP *cannot* be faster than these methods by 2x and log p  times. More specifically, Eq (2) shows T_broadcase_BE/ T_broadcase_LP < 2. This does not provide an upper-bound of T_broadcase_LP and it can be arbitrary worse when comparing with T_broadcase_BE from this inequality. Therefore, instead of showing T_broadcase_BE/ T_broadcase_LP < 2, the authors should state T_broadcase_BE/ T_broadcase_LP > 1 when n approaches infinity. 

- It would be interesting to emphasize more on the differences between designing parallel algorithms on CPU v.s. on GPU to motivate the paper.

[Author Response · Linnan Wang · 30 Dec 2016]
**Revisions**

To Reviewers,

We have improved the readability according to the feedback from Reviewer 3. Please check at the revision.

[Final Decision · Program Chairs · 06 Feb 2017]
**ICLR committee final decision**

The authors propose improvements for the utilization of modern hardware when training using stochastic gradient. However, the reviewers bring up several issues with the paper, including major clarity issues as well as notational issues and some comments about the theory vs. practice.